# An Evaluation of the Impact of an OPEN Stewardship Generated Feedback Intervention on Antibiotic Prescribing among Primary Care Veterinarians in Canada and Israel

**DOI:** 10.3390/ani14040626

**Published:** 2024-02-16

**Authors:** Kamal R. Acharya, Adar Cohen, Gabrielle Brankston, Jean-Paul R. Soucy, Anette Hulth, Sonja Löfmark, John S. Brownstein, Nadav Davidovich, Moriah E. Ellen, David N. Fisman, Jacob Moran-Gilad, Amir Steinman, Derek R. MacFadden, Amy L. Greer

**Affiliations:** 1Department of Population Medicine, University of Guelph, Guelph, ON N1G 2W1, Canada; brankstg@uoguelph.ca; 2Koret School of Veterinary Medicine, The Robert H. Smith Faculty of Agriculture, Food and Environment, The Hebrew University of Jerusalem, Rehovot 7610001, Israel; adarkohen@gmail.com (A.C.); amirst@savion.huji.ac.il (A.S.); 3Division of Epidemiology, Dalla Lana School of Public Health, University of Toronto, Toronto, ON M5T 3M7, Canada; jeanpaul.soucy@mail.utoronto.ca (J.-P.R.S.); david.fisman@utoronto.ca (D.N.F.); 4Public Health Agency of Sweden, 171 82 Stockholm, Sweden; anette.hulth@folkhalsomyndigheten.se (A.H.); sonja.lofmark@folkhalsomyndigheten.se (S.L.); 5Computational Epidemiology Lab, Boston Children’s Hospital, Boston, MA 02115, USA; john.brownstein@childrens.harvard.edu; 6Harvard Medical School, Harvard University, Boston, MA 02115, USA; 7School of Public Health, Faculty of Health Sciences, Ben-Gurion University of the Negev, Beer Sheva 84105, Israel; nadavd@bgu.ac.il (N.D.); giladko@post.bgu.ac.il (J.M.-G.); 8Department of Health Policy and Management, Guilford Glazer Faculty of Business and Management, Ben-Gurion University of the Negev, Beer Sheva 84105, Israel; ellenmo@bgu.ac.il; 9Department of Health Policy and Management, Faculty of Health Sciences, Ben-Gurion University of the Negev, Beer Sheva 84105, Israel; 10Institute for Health Policy, Management and Evaluation, University of Toronto, Toronto, ON M5T 3M6, Canada; 11Ottawa Hospital Research Institute, Ottawa, ON K1Y 4E9, Canada; dmacfadden@toh.ca

**Keywords:** antibiotic stewardship, interrupted time series analysis, antibiotic prescribing, veterinarian

## Abstract

**Simple Summary:**

The use of antibiotics by veterinarians can be reduced by providing them with periodic feedback on their antibiotic use. This strategy was evaluated among veterinarians in Canada and Israel by using an interrupted time-series study design. The veterinarians were provided with three feedback reports at an interval. The report contained a visualization comparing their prescribing with their peers. The report was supplemented with a guideline on antibiotic prescribing. It was found that the antibiotic prescribing of the veterinarians reduced significantly with some strong indication of geographical variation in the reduction in antibiotic prescribing due to the periodic feedback. This strategy is highly recommended in a veterinary clinic to reduce the unnecessary use of antibiotics.

**Abstract:**

An interrupted time-series study design was implemented to evaluate the impact of antibiotic stewardship interventions on antibiotic prescribing among veterinarians. A total of 48 veterinarians were enrolled in Canada and Israel and their prescribing data between 2019 and 2021 were obtained. As an intervention, veterinarians periodically received three feedback reports comprising feedback on the participants’ antibiotic prescribing and prescribing guidelines. A change in the level and trend of antibiotic prescribing after the administration of the intervention was compared using a multi-level generalized linear mixed-effect negative-binomial model. After the receipt of the first (incidence rate ratios [IRR] = 0.88; 95% confidence interval (CI): 0.79, 0.98), and second (IRR = 0.85; 95% CI: 0.75, 0.97) feedback reports, there was a reduced prescribing rate of total antibiotic when other parameters were held constant. This decline was more pronounced among Israeli veterinarians compared to Canadian veterinarians. When other parameters were held constant, the prescribing of critical antibiotics by Canadian veterinarians decreased by a factor of 0.39 compared to that of Israeli veterinarians. Evidently, antibiotic stewardship interventions can improve antibiotic prescribing in a veterinary setting. The strategy to sustain the effect of feedback reports and the determinants of differences between the two cohorts should be further explored.

## 1. Introduction

Antibiotics are used in animals for therapy, prophylaxis, and growth promotion [1]. While most antibiotic use in animal settings is justifiable, their use for prophylaxis, growth promotion, inappropriate use of antibiotics for the treatment of infections due to weak veterinary control of antibiotics, and frequent use of substandard antibiotics requires additional scrutiny [1,2,3,4,5]. These concerns, especially as they relate to the misuse in animals of critically important antibiotics (CIA) for humans, arise from the fact that the use of antibiotics selects for antibiotic resistant organisms (AROs) in animals [6,7,8,9,10,11,12], which can be readily spread to humans and the environment via various routes [7,13,14]. Despite these concerns, in the absence of interventions, global antibiotic use is expected to increase even further [15,16]. The inappropriate use of antibiotics can be reduced, thereby reducing the prevalence of AROs, by the adoption of antimicrobial stewardship (AMS) measures in both animal and human healthcare settings [17,18,19].

Worldwide, AMS programs have been initiated to preserve the effectiveness of antibiotics, reduce the development and prevalence of antimicrobial resistant (AMR) bacteria, and protect human and animal health [17,20,21]. These programs consider various AMS interventions to support behavioural changes in prescribing. Prescribing behaviour is a complex phenomenon influenced by individual, societal, economic, and cultural factors [22]. Broadly, AMS interventions can be (1) prescribing or restrictive when the prescription of antibiotics is either not permitted at all or can only be prescribed upon pre-authorization and justification or (2) persuasive or enabling wherein a prescriber is provided with information and knowledge to make decisions on antibiotic prescribing [23]. In human healthcare settings, both enabling and restrictive antibiotic stewardship interventions have reduced the use of antibiotics [23,24,25] by improving both the duration and amount of antibiotics prescribed [23,26,27]. Prescribing measures such as restriction of the use of antibiotics have been shown to reduce the use of antibiotics in animals thereby decreasing the prevalence of AMR in animals and humans [17,28,29,30]. Widely used measures such as audit and feedback on antibiotic use that have resulted in positive prescribing behaviour change among physicians [31,32,33] may be a practical and sustainable alternative in animal healthcare settings.

In human healthcare settings, the implementation of audit and feedback reporting combined with best practice guidelines has been successful in reducing the use of antibiotics [31,32]. An AMS program requires a mechanism to track, evaluate, and report on antibiotic prescribing patterns by a prescriber, as well as easy access to recent evidence-based prescribing guidelines [34]. The resources and technical tools required for AMS can be barriers to operationalize AMS in veterinary practices [35,36,37]. Therefore, an Online Platform for Expanding Antibiotic Stewardship (OPEN Stewardship) has been developed that can generate periodic individualized feedback reports on antibiotic prescribing by comparing individual prescribers with a group of their peers, as well as providing them with easy access to prescribing guidelines [38]. The validation of this strategy in a veterinary setting will be valuable for supporting the implementation of AMS in veterinary clinics.

In Israel, limited veterinary specific AMS tools are in place. For example, a voluntary survey of antibiotic consumption begun in 2014, and a surveillance of antibiotic resistance among bacteria in slaughterhouses and abattoirs is expected to begin in 2022 [39]. Whereas in Canada, a number of AMS initiatives have been initiated, including surveillance of antibiotic use in animals, stakeholder engagement, publication of guidelines on antibiotic use, restriction on the use of medically important antibiotics in animals, and development and deployment of a digital application supporting the prescribing of antibiotics [40,41,42,43]. However, research evaluating the use of AMS interventions on antibiotic prescribing in the veterinary sector in both Israel and Canada or such evaluation across two countries have not been conducted. This study evaluated the impact of periodic feedback reports on antibiotic prescribing coupled with relevant prescribing guidelines among veterinarians in Canada and Israel on a veterinarian’s (a) CIA prescribing rate, (b) broad-spectrum antibiotic prescribing rate, and (c) total antibiotic prescribing rate. While comparing the impact of stewardship intervention in different geographical settings is not the main purpose of the study, the difference in the level of exposure to stewardship interventions, the size of the countries and context of the veterinary practices are expected to provide additional value in assessing the impact of stewardship intervention on antibiotic prescribing in veterinary practice and hence the adoption of the OPEN Stewardship in veterinary clinics worldwide.

## 2. Materials and Methods

### 2.1. Study Design and Setting

The study was designed as an interrupted time series and conducted between December 2020 and December 2021 according to a previously published protocol [35] with some modifications due to the SARS-CoV-2 pandemic. The purpose of this study was to report the impacts of the intervention on veterinarian prescribing. This study is part of a large study that also evaluated the impact of the AMS intervention among physicians and measured prescriber satisfaction with the AMS feedback, a detailed report on which will be reported separately [44]. In brief, veterinarians from Canada and Israel were voluntarily enrolled in the study in two waves. The prescribing data of participating veterinarians were obtained for three time periods, (1) pre-intervention, (2) during the intervention, and (3) post-intervention. Appendix A shows the pre-intervention and post-intervention period for each cohort of participants in Canada and Israel. The prescribing data were used to generate three feedback reports (interventions) that were provided to the participants 2–3 months apart. Upon receiving the feedback report, the participants also evaluated each report for its usability and usefulness, though these descriptive findings will be reported in a separate publication.

This study was approved by research ethics boards in both Canada (University of Guelph REB #19-03-006), and Israel (Internal Research Review Committee of the Koret School of Veterinary Medicine—Veterinary Teaching Hospital #KSVM-VTH/05_2019). The stakeholders of this study (veterinarians) were involved during both the conception and implementation phases of the study.

### 2.2. Recruitment

In Canada, during the first phase of recruitment (31 January 2020), veterinarians were contacted through their professional associations. Some of the veterinarians that provided consent to be contacted during another study were directly contacted. During the last phase of the recruitment, veterinarians were contacted via national and regional veterinary professional associations, veterinary schools, and veterinary practices. The study invitation was also posted on the social media page of the researchers. In Israel, in the first phase of recruitment (15 March 2019), a phone invitation was extended to clinics in the central and southern districts of Israel. A follow up email was sent on 3 April 2019. Control cohorts were not recruited for the study as the participation rate was strongly influenced by the pandemic. As a result, all of the participants enrolled in the study were assigned to the intervention group. In Canada, participating veterinarians and staff involved in the study were compensated with an electronic gift card of CAD 75 and CAD 30, respectively. However, in Israel, compensation was not provided to the participants in line with local standards.

### 2.3. Intervention

The intervention included the administration of three personalized feedback reports generated by the OPEN Stewardship platform using the prescribing data of the participants. Appendix A shows the timeline of the project with timepoints when each of the three feedback reports were emailed to the participants in Canada and Israel. The first feedback report on critically important antibiotics (CIA) provided participants with a bar graph that compared their prescribing rate of CIA to the mean and the 25th percentile prescribing rate among other study participants located in the same geographical area. Participants were also provided with a guideline document related to the use of CIA that was prepared from the recommendations of the World Health Organization (WHO) and the World Organization for Animal Health [45,46]. The second feedback report on the duration of therapy, provided feedback on the duration of antibiotic prescribing by each participant coupled with a guideline prepared from the available recommendations related to the suggested duration of therapy for registered veterinary products [47,48]. Finally, the third feedback report on the use of broad-spectrum antibiotics focused on the participant’s broad-spectrum antibiotic prescribing rate and provided them with a guideline on using broad-spectrum antibiotics.

### 2.4. Data Extraction and Aggregation

Protocols were tailored to facilitate the extraction of prescribing data from various practice management software systems used by the participating clinics. Data on all medicines prescribed, days worked, and the number of visits per month for each of the participants were obtained on a regular basis over the study period.

The antibacterial products reported by participants were then searched in the online Drug Product Database (DPD) of Health Canada to identify all of the products containing antibacterial active ingredients and the name of the antibacterial component was identified [49]. A compendium of veterinary products, Canada edition, was also consulted for triangulation of the information contained in the DPD. The antibiotics identified through this process were then classified as CIA or other (other than CIA) based on the classification system proposed by the WHO [46]. Antibiotics identified during the study were also classified as broad- or narrow-spectrum based on their spectrum of activity.

The number of prescriptions per month was calculated as the number of antibiotics prescribed by a veterinarian for all of their clients. If a product had more than one antibacterial active ingredients, the number of prescriptions for that product was equal to the number of active ingredients in it. This approach was taken instead of defined daily doses prescribed, as it was not possible to obtain information on the amount of antibiotic prescribed and the demographics of the patients that were treated by each veterinarian in the study. It was assumed that the patient demographics of each veterinarian did not change during the study period, and that the case composition did not vary significantly between practices.

The number of visits per month was calculated as the number of client contacts made during a month. It included both in-person and virtual contacts made by a veterinarian irrespective of whether a prescription was provided to the client. It is possible that a client could contact a veterinarian for more than one animal. In this case, the number of visits was calculated as the number of contacts for each animal. This however could not be established for all veterinary clients as in the case of some herd level visits for large animal practitioners.

The number of days worked was calculated based on the dates in which the clients were invoiced for different reasons, even if the cost was waived by the clinic. In some cases, clinic managers and/or veterinarians were contacted to determine the total number of days worked per month and the total monthly visits. As expected, there could be some variation in the way those dates are recorded by a clinic. Since the data are aggregated by month, this was not likely to impact our analysis or interpretation.

### 2.5. Outcome

The primary outcome of interest was the total number of prescriptions of antibiotics per month per 100 visits. Likewise, the monthly rate of prescribing CIA and broad-spectrum antibiotics per 100 visits were other outcomes of interest.

### 2.6. Statistical Analysis

The impact of the intervention on the prescribing rate of the participants was evaluated using a quasi-experimental interrupted time series design. In the multi-level generalized linear mixed-effect model, both change in the level of antibiotic prescribing as well as change in the trend of antibiotic prescribing following the administration of the intervention was assessed [50]. Pre (0) and post (1) interventions were included as a dummy variable. Figure 1 shows the pre and post intervention period for different cohorts of participants in the study. Change in trend was tested in the model by including each month past the intervention month as a discrete variable. After testing for overdispersion as described [51], negative binomial regression was used to analyse the data [52,53]. Data analysis was conducted in R using the lme4 package [54]. Figures were produced in R using the ggplot2 package [55]. The region was included in the model as a fixed effect as there was expected variance in prescribing rate between the two geographical locations of the study participants. Likewise, participants were nested within region and were included as a random effect in the model to account for the variation in the prescribing rate for individual prescribers within a region. A random residual with an autoregressive variance structure was used to account for correlation of the prescribing rate over time. The number of visits per month was included in the model as an offset term after log transformation to consider the difference in the monthly number of visits for the participants. Interaction between the intervention and time post intervention was also assessed in the model. Likewise, residual autocorrelation was also evaluated during analysis. Both forward stepwise selection and backward stepwise elimination approaches to model building were used. The model estimates were exponentiated to obtain the incidence rate ratio and 95% confidence interval. P-values less than 0.05 were considered to represent a significant difference. To validate the observed effect of the intervention, the model was evaluated by including number of visits as a proxy outcome.

## 3. Results

### 3.1. Study Participants

A total of 21 small animal veterinarians and 12 bovine veterinarians consented to participate in the study in Canada. In Israel, a total of 65 small animal veterinarians from 10 clinics gave their consent to participate in the study. Over the course of the study, a significant amount of participant dropout was observed in both countries. In Canada, most of the participants who dropped out from the study cited reasons including no longer practicing, absence of electronic practice management software, concern over data security, absence of electronic record keeping, difficulty in accessing prescribing data from the practice management software, major transition of the clinic, staff shortages, lack of additional compensation, and transition of the practice management software. Similarly, in Israel, the drop out reasons were concerns over data security, change in the practice management software, closure of the clinic during the study, absence of electronic record keeping, difficulty in accessing prescribing data from the practice management software, and no longer practicing.

In Canada, the final sample size was 10 small animal veterinarians and two bovine veterinarians from 11 clinics. The final sample size in Israel was 36 small animal veterinarians from seven clinics. In total, 48 veterinarians from two countries were enrolled in the study. Appendix A shows the timeline when each cohort of participants from Canada and Israel received interventions.

### 3.2. Antibiotic Prescribing

The study included a total of 14,060 antibiotic prescriptions from the participating veterinarians representing 62,053 visits over 10,442 days of work. Of the total antibiotic prescribing, 8376 prescriptions were for antibiotics considered of critical importance (CIA) for humans. Likewise, considering the spectrum of the antibiotics, a total of 5630 broad-spectrum and 8430 narrow-spectrum antibiotics were prescribed during the study period. The median monthly prescribing of total antibiotics was 18.00 (10.00–31.00), broad-spectrum antibiotics was 7.00 (4.00–13.00) and CIA was 11.00 (6.00–18.00) (Table 1). The total days worked ranged from a single day to 31 days with a median value of 17.00 (13.00–21.00). The median of the total monthly visits was 86.00 (46.25–129.00). At the country level, compared to Israeli veterinarians, the Canadian veterinarians worked for more days with a median value of 21.00 (17.00–25.00) (Table 1). The median number of cases attended by Canadian veterinarians was 131.00 (90.00–216.00) and that of Israeli veterinarians was 70.00 (39.00–109.00) (Table 1). Consequently, the total number of antibiotics prescribed by Canadian veterinarians (mean = 25.10) was higher than that of Israeli veterinarians (mean = 21.78). However, Israeli veterinarians tended to prescribe a greater number of CIA (mean = 14.32) and broad-spectrum antibiotics (mean = 10.30) when compared to their Canadian counterparts who prescribed an average of 11.69 CIA and 6.24 broad-spectrum antibiotics (Table 1).

After receipt of the first feedback report, there was a decline in monthly mean antibiotic prescribing, which was more pronounced among participants in Israel (Figure 1). Likewise, the mean of the total antibiotic prescribing among veterinarians in Canada and Israel decreased immediately after the first, second, and third feedback reports (Figure 2, Figure 3 and Figure 4). However, the decline was less pronounced with each successive feedback report. Similarly, Figure 5, Figure 6 and Figure 7 show the impact of the first, second, and third feedback reports on reducing the prescribing of CIA in both Israel and Canada. The decline was observed to be sharper in Israel compared to that in Canada (Figure 5, Figure 6 and Figure 7). Over the entire study period, the proportion of narrow-spectrum antibiotic prescriptions was higher than that of the broad-spectrum antibiotics (Figure 8). However, after receiving the third feedback report that focused on prescribing less broad-spectrum antibiotics, more broad-spectrum antibiotics were prescribed by the participants (Figure 8).

### 3.3. Multi-Level Generalized Linear Mixed Effect Model

The first two feedback reports resulted in a reduced rate of total antibiotic prescribing with incidence rate ratios (IRR) of 0.88 (0.79, 0.98), and 0.85 (0.75, 0.97). The receipt of the third feedback report led to an increase in total antibiotic prescribing with an IRR of 1.38 (1.23, 1.56) (Table 2). Likewise, when compared to the participants in Israel, veterinarians in Canada were less likely to prescribe antibiotics (IRR = 0.56 (0.41, 0.76) Table 2). Similarly, holding all other parameters constant, the receipt of the first (IRR = 0.87; 0.77, 0.98) and second (IRR = 0.80; 0.70, 0.93) feedback reports subsequently reduced the prescribing rate of critical antibiotics (Table 3). When other parameters were held constant, the rate of prescribing of critical antibiotics by Canadian veterinarians decreased by a factor of 0.39 compared to that of Israeli veterinarians (Table 3). Finally, the rate of prescribing of broad-spectrum antibiotics increased by a factor of 1.48 (95% confidence Interval: 1.28, 1.71) after the receipt of the third feedback report when other variables were held constant (Table 4).

## 4. Discussion

To the best of our knowledge, this is the first study to evaluate the impact of administering a combination of two types of enabling antibiotic stewardship interventions among veterinarians in two countries using a prospective interrupted time series study design [56].

Several indicators have been used to evaluate the success of antibiotic stewardship interventions [57]. In this report, the ability of the stewardship intervention to reduce the use of critically important antibiotics and total antibiotic use was measured. These outcomes were the focus of this report as there is a significant concern regarding the inappropriate use of antibiotics in companion and large animals that are known to be critical for human medicine. For example, a large proportion of critical antibiotics prescribed for companion animals are likely inappropriate and their use can be difficult to justify [46,58]. Similarly, in large animal veterinary medicine, critically important antibiotics are sometimes used for herd-wide therapy for the management of mastitis and without the identification of the causative pathogen [46,59,60,61].

As expected, geographical and individual level heterogeneity in effect size was observed in this study. A more significant change in prescribing patterns was seen among Israeli participants when compared to Canadian participants. This is likely because this was the first stewardship intervention of this type administered to the Israeli veterinarians who historically have had less exposure to antibiotic stewardship measures. Although none of the participating veterinarians in Canada had implemented stewardship interventions in their practices, some stewardship interventions have been investigated previously in Canada [62,63], and even during the time of this study, several concurrent veterinary AMS studies were underway since this is a priority area for research. Likewise, a higher baseline prescribing rate in Israel compared to that in Canada could be another reason for a remarkable effect in Israel as evident from a meta-analysis which suggests that an audit and feedback intervention is more effective when the baseline prescribing is high [31]. Similarly, the pre-intervention prescribing rate of veterinarians that was used as a control to assess the post-intervention prescribing rate of the participants was from both the pre- and post-pandemic period in Israel (discontinuous) and only post-pandemic period in Canada (continuous). This might have some impact in the effect which is unknown.

While the effect of the antibiotic stewardship intervention was sustained during the study period, after the last feedback report there was a rebound in the use of antibiotics that was almost as high as the pre-intervention level. Had it been feasible, a qualitative investigation among selected veterinarians could have been performed to better understand this increase. It might be that instead of clearly discouraging the prescription of broad-spectrum antibiotics, veterinarians were reminded of their existence, and thus placing them on top of their minds. It shows that guidelines must be set up in a clear way and different presentations should be more widely tested among users. Additionally, there is a possibility that at the end of the study period the last feedback report might not have been received by the study participants as intended. Combined with the ongoing challenges encountered by the veterinarians as a result of the pandemic this may have contributed to our findings [64]. The study team also had to follow-up with participants more frequently after the last feedback report to encourage receipt and evaluation of the third feedback report. Another possible explanation for this occurrence could be related to changes in veterinary practices and case composition in both study locations during the study period due to public health measures to manage the SARS-CoV-2 pandemic. The first stay-at-home order due to the COVID-19 pandemic led to a drastic change in the way veterinary clinics operated in Canada. Non-essential procedures were not performed, and telemedicine and online consultation were used by most clinics as the primary mode of service delivery. Likewise, animals were picked up at curb side and returned to the owners after the procedure which could have demotivated an owner to access veterinary service [65]. Access to clinics could also have been impacted due to a clients’ changed financial situation due to the pandemic [65]. As a result, the case composition in the participating clinics was expected to have changed in favour of cases that likely required serious medical attention although veterinarians’ days of work and case attended were comparable to the pre-study level (Appendix A). With the relaxation of some public health restrictions, the case composition was expected to have changed further in Canada. In the case of Israel, veterinary clinics remained open as usual as veterinary care was deemed an essential service, and it has been noted anecdotally that there may have been an increase in case load possibly driven by the increase in adoption of animals during the pandemic and added attention of the owners towards their pets. However, among the study participants in Israel, for fewer days worked there were a comparable number of visits attended.

The voluntary nature of the study may have engaged participants who had previous experience with antibiotic stewardship and/or those that were keen on the institutionalization of antibiotic stewardship interventions. Consequently, the effect size could be different for participants that are less interested in antibiotic stewardship. Nevertheless, veterinarians are expected to use antibiotics responsibly. So, the effect size of this intervention is expected to be larger and sustained longer than what is evident in this study, especially when the practice management and peers are involved in the process and specific targets are provided to the prescribers [31]. The study also identified some clinic level and individual level reasons why participants dropped out of the study. Many of these concerns could have been addressed by visiting the clinics, which was not possible due to the pandemic restrictions in place.

Due to the pandemic, recruitment and retention was challenging, however a minimum sample size for adequate statistical power was reached. For similar reasons, parallel control participants were not included in the study. Similarly, the veterinary clinics in the study varied in their specialty and the level of care they provided. This could have introduced heterogeneity in the effect size. The researchers could not visit the clinics to facilitate the data extraction. While some practices were able to proceed with telephone and video call support to get the data extracted, some practices were not able to do so which impacted the granularity of the prescribing data. For some practices, it was impossible to obtain information on the duration of prescribing of an antibiotic, demographics of the animals, tentative or definitive diagnosis made on a case, and/or the dose of antibiotic prescribed. This limited us from using more universal measures of antibiotic use such as defined daily dose and duration of therapy. However, the unit used in this study (number of prescriptions of antibiotics per 100 visits) is a simple and intuitive measure of antibiotic use best suited to the scenario where the diversity of practice management software being used in veterinary practices makes it difficult to extract the data that is required to compute epidemiologically useful but hard to interpret measures such as defined daily dose [66,67]. Likewise, in addition to the observation period being insufficient to assess the seasonality of antibiotic prescribing [50], the pandemic probably disrupted any of the seasonal prescribing trends. The use of real-time data to generate the feedback report would be a motivator for the prescribers who changed their prescribing patterns due to the feedback reports. This would also encourage prescribers who were less affected by the feedback report to reflect on their practices as the improvement in prescribing of other participants would make their prescribing look less appropriate. However, due to the change in veterinary practice during the pandemic, it was impossible to obtain data in real-time and use them to prepare feedback reports to the participants. Instead, prescribing data before the study duration was used to generate the feedback report. Therefore, caution should be taken while interpreting the results of this study for different types of veterinary practices.

The inclusion of two geographical locations was a strength of this study. The participating veterinarians in the study showed a huge range in their years of experience. For example, the year of graduation of the participating Canadian veterinarians ranged from 1985 to 2019. While, similar data was not obtained from Israel, the variation in the age (32 to 57 years) of the participating Israeli veterinarians suggests a similar range. Likewise, the study was able to develop ways to utilize prescribing data obtained from diverse practice management software being used in veterinary clinics such that the automation of feedback reports using OPEN stewardship would be possible for many practice management tools in use.

## 5. Conclusions

Periodic administration of feedback reports on antibiotic prescribing combined with prescribing guidelines improved the prescribing of antibiotics among veterinarians. Although there were variations in the prescribing patterns over the intervention period and in the two countries overall, veterinarians prescribed fewer broad-spectrum antibiotics and fewer antibiotics of critical importance to humans because of the intervention study. The use of simple metrics of antibiotic prescribing made this approach intuitive as well as adaptable to diverse veterinary practices in two very distant locations in the world. For maintenance of the effect, the feedback reports should be routinely provided to the veterinarians preferably from management or their peers as suggested by previous studies [31]. Further work should focus on measuring the effect size in different types of veterinary practices using appropriate sample size, making comparisons with control participants, and following the participants through a longer post intervention period.

## Figures and Tables

**Figure 1 animals-14-00626-f001:**
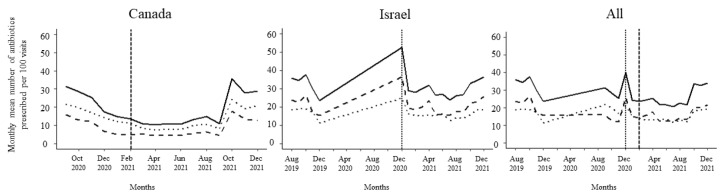
Monthly mean number of antibiotics prescribed per 100 visits by the participating veterinarians in Canada, Israel and both study sites. Solid line represents total antimicrobials prescribing, dashed line represents critical antimicrobials prescribing, and dotted line represent narrow-spectrum antimicrobials prescribing. Dashed and dotted vertical reference lines represent the months in which interventions were administered for the first time to the study participants in Canada and Israel.

**Figure 2 animals-14-00626-f002:**
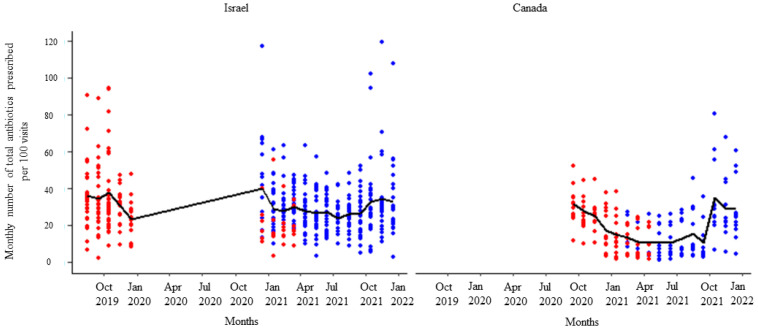
Number of total antibiotic prescriptions per month per 100 visits by the participating veterinarians before (red dots) and after (blue dots) the first feedback report. The solid black line represents the monthly mean number of total antimicrobials prescribed.

**Figure 3 animals-14-00626-f003:**
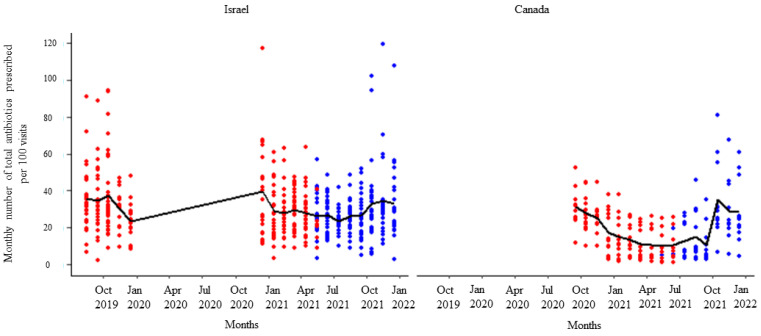
Number of total antibiotic prescriptions per month per 100 visits by the participating veterinarians before (red dots) and after (blue dots) the second feedback report. The solid black line represents the monthly mean number of total antimicrobials prescribed.

**Figure 4 animals-14-00626-f004:**
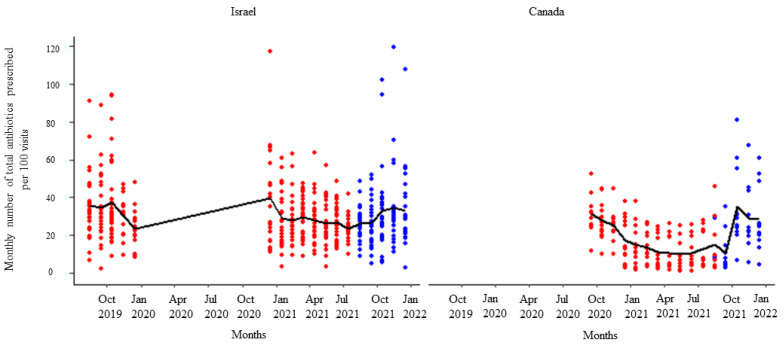
Number of total antibiotic prescriptions per month per 100 visits by the participating veterinarians before (red dots) and after (blue dots) the third feedback report. The solid black line represents the monthly mean number of total antimicrobials prescribed.

**Figure 5 animals-14-00626-f005:**
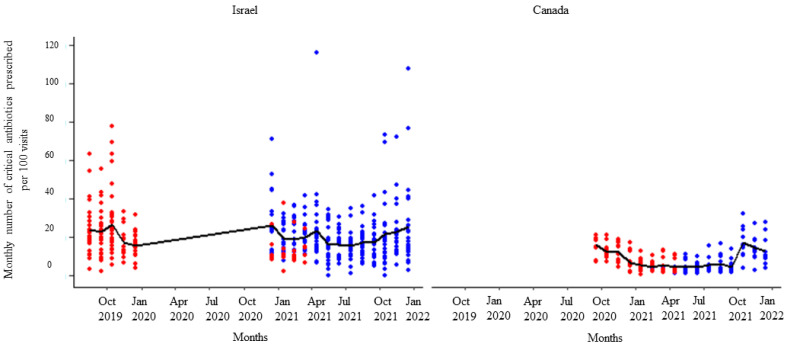
Number of critical antibiotic prescriptions per month per 100 visits by the participating veterinarians before (red dots) and after (blue dots) the first feedback report. The solid black line represents the monthly mean number of critical antimicrobials prescribed.

**Figure 6 animals-14-00626-f006:**
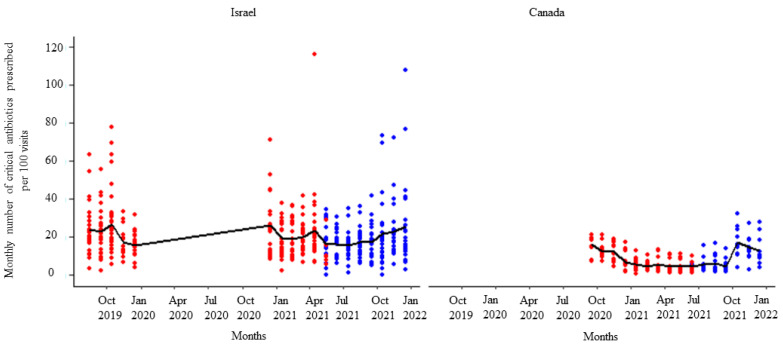
Number of critical antibiotic prescriptions per month per 100 visits by the participating veterinarians before (red dots) and after (blue dots) the second feedback report. The solid black line represents the monthly mean number of critical antimicrobials prescribed.

**Figure 7 animals-14-00626-f007:**
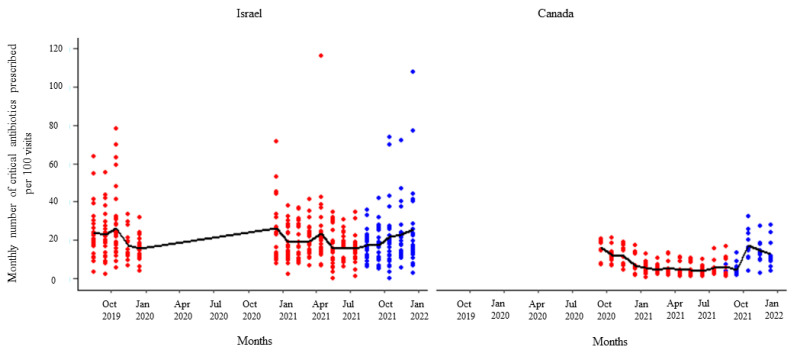
Number of critical antibiotic prescriptions per month per 100 visits by the participating veterinarians before (red sots) and after (blue dots) the third feedback report. The solid black line represents the monthly mean number of critical antimicrobials prescribed.

**Figure 8 animals-14-00626-f008:**
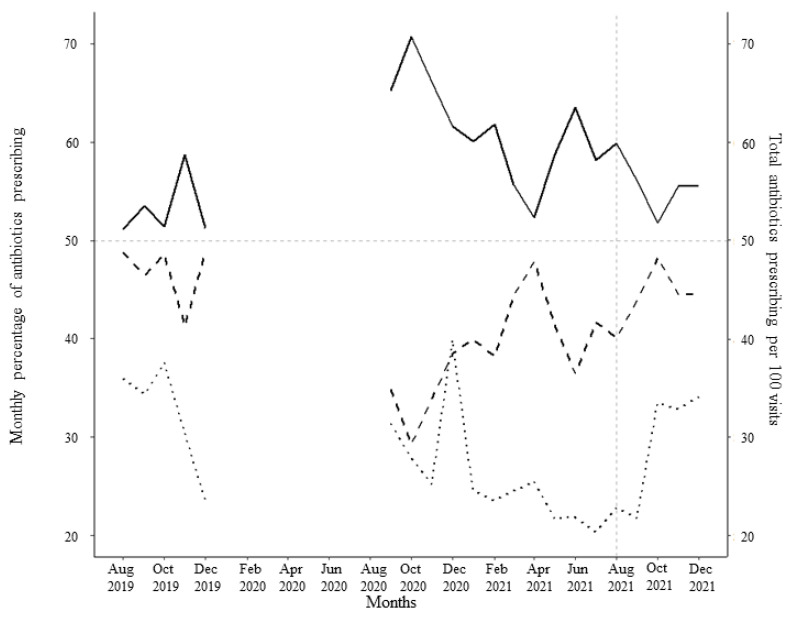
Mean monthly percentage of narrow-spectrum antibiotics (solid line) and broad-spectrum antibiotics (dashed line) prescribed by the participating veterinarians before and after the third feedback report on August 2021 (grey vertical dashed line). The horizontal grey dashed line represents 50% of antimicrobial prescribing. The dotted line represents the total antimicrobial prescriptions per 100 visits during the time period.

**Table 1 animals-14-00626-t001:** Summary statistics of the veterinarians’ monthly antibiotics prescribing, days worked and total visits during the study period stratified by regions (Israel and Canada) *.

Region	Summary Statistics	Monthly Antibiotics Prescribing					Days Worked per Month	Total Visits per Month
		Critical	Other	Total	Broad-Spectrum	Narrow-Spectrum		
Israel	Min	0.00	0.00	1.00	0.00	0.00	1.00	5.00
	First quartile	7.00	3.00	11.00	4.00	5.00	12.00	39.00
	median	12.00	6.00	18.00	8.00	10.00	16.00	70.00
	Mean	14.32	7.45	21.78	10.30	11.48	15.35	77.00
	Third quartile	19.00	10.00	30.00	14.00	16.00	19.00	109.00
	Max	78.00	34.00	98.00	78.00	64.00	26.00	225.00
Canada	Min	1.00	0.00	3.00	0.00	1.00	6.00	24.00
	First quartile	4.00	4.00	9.00	2.00	5.00	17.00	90.00
	median	7.00	8.00	18.00	4.00	10.00	21.00	131.00
	Mean	11.69	13.41	25.10	6.24	18.86	20.64	156.92
	Third quartile	16.00	16.00	35.00	9.00	24.00	25.00	216.00
	Max	103.00	68.00	160.00	29.00	139.00	31.00	488.00

* More about the overall prescribing scenario of all participants can be found in Appendix A.

**Table 2 animals-14-00626-t002:** Effect of three feedback reports and geography on total antibiotics prescribing.

Fixed Effects	Incidence Rate Ratio (95% Confidence Interval)	*p*-Value
Intercept	0.32 (0.27, 0.37)	<0.01
Region		<0.01
Israel	1.00	
Canada	0.56 (0.41, 0.76)	
Feedback 1	0.88 (0.79, 0.98)	0.02
Feedback 2	0.85 (0.75, 0.97)	0.01
Feedback 3	1.38 (1.23, 1.56)	<0.01

Mixed effects generalized linear regression model equation: total antibiotics prescribing~ 1 + region + First feedback report + Second feedback report + Third feedback report + (1 | Region: Participant ID).

**Table 3 animals-14-00626-t003:** Effect of three feedback reports and geography on critical antibiotics prescribing.

Fixed Effects	Incidence Rate Ratio (95% Confidence Interval)	*p*-Value
Intercept	0.21 (0.18, 0.24)	<0.01
Region		<0.01
Israel	1.00	
Canada	0.39 (0.29, 0.52)	
Feedback 1	0.87 (0.77, 0.98)	0.02
Feedback 2	0.80 (0.70, 0.93)	<0.01
Feedback 3	1.49 (1.30, 1.70)	<0.01

Mixed effects generalized linear regression model equation: Critical antibiotics prescribing~ 1 + region + First feedback report + Second feedback report + Third feedback report + (1 | Region: Participant ID).

**Table 4 animals-14-00626-t004:** Effect of three feedback reports and geography on broad-spectrum antibiotics prescribing.

Fixed Effects	Incidence Rate Ratio (95% Confidence Interval)	*p*-Value
Intercept	0.15 (0.12, 0.18)	<0.01
Region		<0.01
Israel	1.00	
Canada	0.30 (0.21, 0.43)	
Feedback 1	0.86 (0.76, 0.98)	0.02
Feedback 2	0.81 (0.69, 0.94)	0.01
Feedback 3	1.48 (1.28, 1.71)	<0.01

Mixed effects generalized linear regression model equation: Broad-spectrum antibiotics prescribing~ 1 + Region + First feedback report + Second feedback report + Third feedback report + (1 | Region: Participant ID).

## Data Availability

Data on participating veterinarians and their patients and prescriptions will not be made available for privacy reasons. However, the deidentified data can be obtained on request from the corresponding author.

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
