# Peer review of "An Evaluation of the Impact of an OPEN Stewardship Generated Feedback Intervention on Antibiotic Prescribing among Primary Care Veterinarians in Canada and Israel"

_animals, 2024, doi:10.3390/ani14040626_

Round 1

Reviewer 1 Report

Comments and Suggestions for Authors

Salami slicing research

Author Response

Dear Reviewer,

We highly appreciate your time in reviewing our paper but we disagree when our work was labelled as "salami slicing  research". While the protocol of this study was published previously before we started this study, this submission is the only instance the finding of the study is being considered for publication and there is no future intention to conduct any further analysis of the data and produce the findings.

Reviewer 2 Report

Comments and Suggestions for Authors

Although the work addresses an important and timely topic, investigating the influence of stewardship-generated feedback intervention on antibiotic prescribing among primary care veterinarians in Canada and Israel, certain considerations should be made before accepting this paper.

1- The authors should explain why they chose to compare Canada and Israel in this study. Is there any connection between the two nations? Why did they not choose two neighboring American or Asian countries?

2- In response to the previous question, the authors highlight in the introduction (L90-L101) that there are differences between Canada and Israel in Stewardship programs, so the authors must explain why they chose both countries and, if possible, provide additional information if any previous comparisons between both countries were performed, even in the human clinical field.

3. Are there been past comparisons between countries in the veterinary or human fields? Please emphasize.

4- Please adjust the figures because their quality is very low and cannot be easily checked. Most crucially, there were far too many figures given. Please cut down on the quantity of figures.

5- What is the difference between Table 1 and Table 2?

6- The discussion should be modified, because the authors just spoke generically and did not go into detail about their findings. 

Comments on the Quality of English Language

I think English quality is acceptable.

Author Response

Point-to-point response to the comments of reviewer

We would like to acknowledge the editor and reviewer for their time in reviewing our manuscript “An interrupted time series to evaluate the impact of an OPEN Stewardship generated feedback intervention on antibiotic prescribing among primary care veterinarians in Canada and Israel” and providing us with their valuable feedbacks. We have provided the point-to-point response to each comment of the reviewer and made respective revision of the manuscript.

Response to the Editor's comments

Although the work addresses an important and timely topic, investigating the influence of stewardship-generated feedback intervention on antibiotic prescribing among primary care veterinarians in Canada and Israel, certain considerations should be made before accepting this paper.

1- The authors should explain why they chose to compare Canada and Israel in this study. Is there any connection between the two nations? Why did they not choose two neighboring American or Asian countries?

Response: The work is a collaboration between researchers from several countries including Canada and Israel where veterinarians were enrolled in the study. Another study from the same collaboration measures the impact in healthcare setting in these countries. We chose these countries to evaluate the impact of stewardship interventions in different settings to validate the tool (OPEN Stewardship). We have added followings: “Apart from the difference in the level of exposure to stewardship interventions, the size of the countries and context of the veterinary practices are expected to provide additional value to the assessment of the impact of stewardship intervention on antibiotic prescribing and hence adoption of the OPEN Stewardship in veterinary clinics worldwide.” ( Line 102 to 107)

2- In response to the previous question, the authors highlight in the introduction (L90- L101) that there are differences between Canada and Israel in Stewardship programs, so the authors must explain why they chose both countries and, if possible, provide additional information if any previous comparisons between both countries were performed, even in the human clinical field.

Response: The changes made in lines 96 to 99 and lines 102 to 107 will make this clearer as:

“However, research evaluating the use of AMS interventions on antibiotic prescribing in the veterinary sector in both Israel and Canada or such evaluation across two countries are largely lacking.”

“Apart from the difference in the level of exposure to stewardship interventions, the size of the countries and context of the veterinary practices are expected to provide additional value to the assessment of the impact of stewardship intervention on antibiotic prescribing and hence adoption of the OPEN Stewardship in veterinary clinics worldwide.”

  1. Are there been past comparisons between countries in the veterinary or human fields? Please emphasize.

Response: To our knowledge, there have not been any study comparing the impact of stewardship interventions between countries. This has been emphasized in line 96-99 as: “However, research evaluating the use of AMS interventions on antibiotic prescribing in the veterinary sector in both Israel and Canada or such evaluation across two countries are largely lacking.”

4- Please adjust the figures because their quality is very low and cannot be easily checked. Most crucially, there were far too many figures given. Please cut down on the quantity of figures. 

Response: The figure quality has been enhanced. Additionally separate figures have been uploaded to the journal’s submission system.

5- What is the difference between Table 1 and Table 2?

Response: Thank you for raising this question. The first table is a summary statistics of monthly antibiotics prescribing, days worked and total visits during the study period of all participating veterinarians across the study while the second is the same information stratified by country. Now we have moved the first one to the supplementary section and retained the second table only in the main manuscript. (see line 265 to 268).

6- The discussion should be modified, because the authors just spoke generically and did not go into detail about their findings.

Response: The discussion is more generic and only focusses on the study as we this is the first study of this kind, and our intention was to provide many details around the study context which would help the readers in interpreting the study findings and future researchers in comparing their study findings with this study.

Round 2

Reviewer 1 Report

Comments and Suggestions for Authors

For the manuscript, it is very important that the wording of the title and adequate methodological explanations avoid the effect of a salami slice publication. The manuscript is well conceived. Check references for MDPI style and possible typos.

Author Response

We appreciate your comment and have provided some detail and reference to other studies that formed a part of the bigger project. However, what we have present is a unique and fulsome study that contributed to the bigger project and hence required a manuscript on its own. Likewise, it would not be reasonable to combine this study findings with the other studies. See: “This study is part of a large study which also evaluated the impact of the AMS intervention among physicians and measured prescriber satisfaction with the AMS feedback, a detailed report on which will be reported separately (47)”,(Lines 115-118).

The manuscript has been thoroughly checked for typos and the references has been checked for MDPI style.

Reviewer 2 Report

Comments and Suggestions for Authors

I don't have more comments, I think the paper is now improved.

Comments on the Quality of English Language

English in OK.

Author Response

Thank you.